# Vesicular-Bound HLA-G as a Predictive Marker for Disease Progression in Epithelial Ovarian Cancer

**DOI:** 10.3390/cancers11081106

**Published:** 2019-08-02

**Authors:** Esther Schwich, Vera Rebmann, Peter A. Horn, Alexander A. Celik, Christina Bade-Döding, Rainer Kimmig, Sabine Kasimir-Bauer, Paul Buderath

**Affiliations:** 1Institute for Transfusion Medicine, University Hospital Essen, University of Duisburg-Essen, Virchowstr. 179, 45147 Essen, Germany; 2Institute for Transfusion Medicine, Hannover Medical School, Carl-Neuberg-Str. 1, 30625 Hannover, Germany; 3Department for Gynecology and Obstetrics, University Hospital Essen, University of Duisburg-Essen, Hufelandstr. 55, 45147 Essen, Germany

**Keywords:** extracellular vesicles, vesicular-bound HLA-G, HLA-G_EV_, epithelial ovarian cancer (EOC), liquid biopsy, platinum therapy, residual tumor burden, circulating tumor cells

## Abstract

Extracellular vesicles (EV) and their tumor-supporting cargos provide a promising translational potential in liquid biopsies for risk assessment of epithelial ovarian cancer (EOC) patients frequently relapsing, despite initial complete therapy responses. As the immune checkpoint molecule HLA-G, which is operative in immune-escape, can be released by EV, we evaluate the abundance of EV and its vesicular-bound amount of HLA-G (HLA-G_EV_) as a biomarker in EOC. After enrichment of EV from plasma samples, we determined the EV particle number and amount of HLA-G_EV_ by nanoparticle tracking analysis or ELISA. The association of results with the clinical status/outcome revealed that both, EV particle number and HLA-G_EV_ were significantly elevated in EOC patients, compared to healthy females. However, elevated levels of HLA-G_EV_, but not EV numbers, were exclusively associated with a disadvantageous clinical status/outcome, including residual tumor, presence of circulating tumor cells, and disease progression. High HLA-G_EV_ status was an independent predictor of progression, besides residual tumor burden and platinum-sensitivity. Especially among patients without residual tumor burden or with platinum-sensitivity, HLA-G_EV_ identified patients with high risk of progression. Thus, this study highlights HLA-G_EV_ as a potential novel biomarker for risk assessment of EOC patients with a rather beneficial prognosis defined by platinum-sensitivity or lack of residual tumor burden.

## 1. Introduction

Epithelial ovarian cancer (EOC) is the most lethal gynecologic malignancy [1]. According to the International Federation of Gynecology and Obstetrics (FIGO), it is classified into four surgical stages considering the extent of ovary affection and the extent of spreading outside the ovaries and outside the pelvis [1]. Primary treatment of EOC consists of cytoreductive surgery followed by adjuvant, platinum-based chemotherapy. Although initially responding to the primary treatment, tumor regrowth with a drug-resistant phenotype is frequent, resulting in an impaired overall survival [2]. Biomarkers predicting therapy response and prognosis in EOC are, therefore, desperately needed. In this context, circulating tumor cells (CTC) in the peripheral blood of EOC patients have been studied extensively, and their prognostic significance has been clearly shown [3]. However, although CTC have been well characterized in EOC patients [4,5], it seems that they preferentially interact with other components of their environment to promote metastases than being solely responsible for metastasis formation. In breast cancer, we recently demonstrated a relationship, as well as a prognostic impact of CTC and circulating extracellular vesicles (EV) after neoadjuvant chemotherapy [6].

EV are membrane-surrounded structures released into the extracellular space by both healthy cells (e.g., trophoblasts and adult and embryonic stem cells [7]) and transformed cells, including breast cancer cells [6,8], melanoma [9,10], and renal cancer cells [11,12]. The generic term EV comprises lipoproteic vesicles of different sizes, origin, and composition, such as exosomes, microvesicles, and apoptotic bodies, making EV a very heterogeneous population [10,13]. EV are transported outside the producing cell and carry proteins, lipids, and nucleic acids characteristic of their cell of origin, being present either on the vesicle membrane or inside the vesicles [14]. As important cell–cell-communication mediators, they have been discussed to promote tumor initiation, metastasis formation, and therapy-resistance in cancer cells [15]. Thus, in the era of liquid biopsy, EV bear great translational potential for disease monitoring and prediction of therapy response and outcome in various malignancies [16,17,18,19,20].

Of note, high levels of EV [6] and high levels of a EV sub-population expressing the non-classical human leukocyte antigen-G (HLA-G) [21] have been associated with the failure of neoadjuvant chemotherapy and disease-progression in peripheral blood of locally advanced, primary breast cancer patients.

HLA-G is an immune checkpoint molecule regulating immune effector responses. It interacts with the immune-inhibitory receptors immunoglobulin-like transcript- (ILT-)2, ILT-4, and killer-cell immunoglobulin-like receptor (KIR)2DL4, expressed on different immune-competent cells [22]. Thus, HLA-G is associated with anti-inflammatory and immune-modulatory properties. HLA-G can be expressed at the cell surface (HLA-G1, -G2, -G3, and -G4) [23] or as soluble HLA-G forms, including secreted molecules (HLA-G5, -G6, and -G7) [24], shed molecules from the surface [25], or as released molecules by extracellular vesicles [22,26]. HLA-G is confined to the maternal–fetal interface and to immune-privileged adult tissues under physiological conditions [27], whereas neo-ectopic or aberrant expression of HLA-G and its soluble forms have been associated with a variety of pathological situations [28]. For instance, HLA-G and its soluble forms enable escape from host immune surveillance by inhibiting B cells, T cells, and natural killer cells and by inducing regulatory T cells [28,29,30], thereby mediating cancer invasiveness and metastatic progression [31]. There is evidence that HLA-G-bearing EV are an especially crucial factor in immune-tolerance mechanisms operative in malignant diseases [21,22], as the total amount of soluble HLA-G has not been associated with disease progression and overall survival in breast cancer patients.

With respect to EOC, the prognostic significance of HLA-G and soluble HLA-G expression is controversially debated, especially in view of a recent study demonstrating elevated soluble HLA-G levels and high HLA-G tissue expression being associated with an improved prognosis [32]. Our group could confirm increased levels of total soluble HLA-G, however, this was not associated with the nodal status, metastasis formation, presence of circulating or disseminated tumor cells prior to therapy and overall (OS), and progression-free survival (PFS) [33].

So far, the clinical importance of EV and of HLA-G-bearing EV in EOC remains unclear. In this retrospective study, we, therefore, enriched EV derived from plasma samples of primary, serous EOC patients by ExoQuick™ precipitation [6,21] and quantified the EV preparations (i) for amount of total EV particle number, (ii) amount of vesicular-bound HLA-G (HLA-G_EV_), and (iii) related the results to clinical parameters, presence of CTCs, and disease outcome.

## 2. Results

### 2.1. Characterization of Extracellular Vesicles in Plasma of EOC Patients and Healthy Controls

EV preparations were analyzed for the presence of typical EV markers, including tetraspanin CD63, TSG101, and Syntenin, as well as the absence of Cytochrome C, as previously recommended [34] as a minimal requirement for the definition of EV, by SDS-PAGE and specific immunoblotting [6,21] for four healthy controls (HC) and four EOC patients (Figure 1A). All preparations showed the typical EV marker profile. Additionally, NTA measurements showed a size distribution (mean ± SD nm) of 125.6 ± 9.2 for EV detected in EOC patients and 123.5 ± 8.1 for HC (Figure 1B), which corresponds to the known size of EV, ranging between 30–150 nm [13]. Both the EV marker analysis and the NTA results indicate that ExoQuick™ precipitation successfully enriched EV from the plasma samples.

### 2.2. Elevated Levels of Extracellular Vesicles in Serous EOC Patients

Levels of EV (median [range] 10^9^/mL) were determined in plasma samples from EOC patients (*n* = 70) and compared to those of HC (*n* = 30) by NTA (Figure 1C). EV particle concentration in EOC samples (1100 [280–3200]) was more than 10-fold elevated, compared to HC ((106.8 [14.3–289]), *p* < 0.0001). Stratification of patients according to FIGO staging revealed that EV levels were significantly elevated compared to HC (FIGO I–II ((930 [450–3200]), *p* = 0.0001; FIGO III (1050 [280–2500]), *p* < 0.0001; FIGO IV (1200 [480–3100]), *p* < 0.0001) independent of the extent of disease (Figure 1D). Further, EV levels were significantly elevated in both, patients without CTC before therapy ((1150 [280–3200]), *p* < 0.0001) and patients with CTC before therapy ((1100 [610–3100]), *p* < 0.0001), compared to HC (Figure 1E), whereas the particle number did not differ between patients with and without CTC. Regarding CTC-subtypes, EV particle number was neither associated with EpCAM positive CTC (EpCAM^+^: 1100 [610-3100] vs. EpCAM^−^: 1150 [280-3200]), nor with MUC positive CTC (MUC^+^: 1100 [750-1600] vs. MUC^−^: 1200 [280-3200]).

### 2.3. Increased Levels of HLA-G_EV_ in Serous EOC Patients

To further characterize the EV in EOC patients, we quantified HLA-G_EV_ (median [range] ng/mL; Table 1) in plasma samples of serous EOC patients (*n* = 78) and compared the amounts to the HLA-G_EV_ levels of 30 HC. Consistent with the total particle concentration, the median HLA-G_EV_ levels of EOC patients (14.3 [2.9–60.4]) were more than 7-fold increased, compared to HC (1.9 [0.0–25.0]; see Figure 2A). Of note, in 6 out of 30 (20%) HC, no HLA-G_EV_ could be detected, whereas all EOC patients revealed substantial amounts of detectable HLA-G_EV_ (*p* < 0.00001). However, particle concentrations did not correlate with HLA-G_EV_ levels (r = −0.06; *p* = 0.507, *n* = 69).

### 2.4. Elevated Levels of HLA-G_EV_ Associate with a Detrimental Clinical Profile of Serous EOC Patients

Regarding FIGO staging, HLA-G_EV_ levels in EOC patients with early FIGO staging (FIGO I–II) were not significantly elevated (13.3 [2.9–20.5]), compared to HC. However, HLA-G_EV_ levels of patients with advanced FIGO stages (FIGO III: 13.1 [3.4–60.4] and FIGO IV: 19.5 [4.5–44.9]) were significantly (*p* < 0.0001) elevated (Figure 2B), compared to HC.

Due to the tumor-supporting features of HLA-G, we further analyzed the association of HLA-G_EV_ with the presence of CTCs before therapy (Figure 2C). Compared to HC, HLA-G_EV_ levels were increased in patients with CTC (29.1 [10.4–44.9]; *p* < 0.0001) and in patients without CTC (13.1 [2.9–60.4], *p* < 0.0001). CTC positive patients showed more than 2-fold higher amounts of vesicular-bound HLA-G (*p* = 0.06) than CTC negative ones.

Concerning CTC specificity, HLA-G_EV_ was significantly (*p* = 0.03) higher in patients with MUC positive CTC (39.6 [12.2–44.9]), compared to patients with MUC negative CTC (13.6 [2.9–60.4]; Figure 3A) with a 3-fold increase in the median HLA-G_EV_ levels. Similar, high HLA-G_EV_ levels were associated (*p* = 0.05) with EpCAM positive CTC (32.7 [15.8–44.9]), compared to EpCAM negative CTC (13.3 [2.9–60.4]; Figure 3B).

Considering residual tumor burden, levels of HLA-G_EV_ were significantly (*p* = 0.03) elevated (18.7 [4.5–60.4]) in patients with a detectable residual tumor after primary surgery, compared to patients without a residual tumor burden (12.7 [2.9–43.7]; Figure 3C).

### 2.5. HLA-G_EV_ Status as a Prognostic Marker for the Prediction of Disease Progression in EOC Patients

Receiver operating curve analysis was used to define the optimal cut-off value for risk assessment regarding disease progression in EOC patients. Using the obtained cut-off value of 18.45 ng/mL HLA-G_EV_ (sensitivity: 46.9%, specificity: 77.8%), *n* = 30 patients were identified with levels higher than 18.45 ng/mL, whereas *n* = 48 patients had lower levels. Kaplan–Meier curve analysis combined with log-rank test showed that EOC patients with HLA-G_EV_ status > 18.45 ng/mL had a significantly (*p* = 0.029) reduced 3-year progression-free survival (PFS) with a median PFS time of 15 ± 2.5 months (HR: 1.8, 95% CI: 1.1–3.6), compared to EOC patients with a low HLA-G_EV_ status with a median PFS time of 29 ± 7.1 months (Figure 4A).

Multivariate analysis, including metastasis and nodal status, presence of CTC before therapy, residual tumor burden, platinum therapy sensitivity, and HLA-G_EV_ level status, was performed. Strikingly, platinum therapy sensitivity (*p* < 0.0001 and *p* < 0.0001), residual tumor burden (*p* = 0.023 and *p* = 0.011), and elevated HLA-G_EV_ (*p* = 0.029 and *p* = 0.006) were shown to be independent prognostic factors for both 3-year and 10-year PFS, respectively, whereas presence of CTC before therapy did not reach significance (*p* = 0.101 and *p* = 0.08; Table 2). Analogous univariate and multivariate evaluations of the OS revealed no association with HLA-G_EV_ status.

As response to platinum therapy and the residual tumor burden are critical factors for the prediction of disease progression, we further stratified the patients according to these parameters. For patients sensitive to platinum therapy (*n* = 51), Kaplan–Meier analysis showed that high HLA-G_EV_ status was significantly (*p* = 0.027) associated with a reduced 3-year PFS, with a median PFS of 23 ± 9 months (HR: 2.3, 95% CI: 1.1–6.5), compared to a low level status with an undefined PFS time (Figure 4B). Similarly, EOC patients with no residual tumor burden (*n* = 37) and high levels status (*n* = 9) exhibited a reduced 3-year PFS (HR: 2.5, 95% CI: 1.1–9.3, *p* = 0.04, Appendix A).

For the long-term observation of disease progression of patients with no residual tumor burden, it became evident that a high status was significantly (*p* = 0.018) associated with an inferior 10-year PFS, with a median survival time of 27 ± 13.4 month (HR: 2.7, 95% CI: 1.3–10.1; Figure 5), compared to a low status of vesicular-bound HLA-G with a median survival time of 68 ± 25.6 months. Of note, all patients with high HLA-G_EV_ (*n* = 9) status experienced disease progression within approximately 5 years after primary diagnosis.

Concerning patients resistant to platinum therapy or patients with a residual tumor, an association with HLA-G_EV_ levels was found in neither of the univariate or multivariate analyses. Thus, HLA-G_EV_ status seems to be a promising factor for the prediction of disease recurrence, in particular for patients considered as low risk patients for disease progression due to their sensitivity to platinum therapy and lack of residual tumor.

## 3. Discussion

EV provide a promising translational potential in the context of malignancies. Both the rate of EV release, as well as the EV-mediated transfer of different cargos, represent novel strategies for disease monitoring [20]. Although the immune checkpoint molecule HLA-G has been associated with cancer invasiveness and metastatic spread in various malignant diseases [35,36], we were not able to correlate the total amount of sHLA-G to the clinical status or outcome of EOC patients in a previous study [33]. Of note, sHLA-G can also be released by EV, which has been proposed to play an important role in the immune-tolerance mechanisms in different malignant diseases [22]. Thus, we hypothesized that overall EV abundance, as well as amounts of vesicular-bound HLA-G, reflects the disease status and outcome of EOC patients. In our study, we could demonstrate that (i) levels of plasma EV in EOC patients were elevated more than 10-fold, compared to HC, (ii) levels of HLA-G_EV_ were more than 7-fold increased in EOC patients, compared to HC, (iii) elevated HLA-G_EV_ levels were associated with an inferior clinical status and outcome regarding residual tumor, presence of CTC, and PFS, that (iv) HLA-G_EV_ status served as an independent marker for risk assessment of disease progression, and, lastly, (v) among patients without residual tumor burden or with platinum-sensitivity, HLA-G_EV_ identified patients with a high risk of progression.

The characterization of EV enriched from EOC patient samples was performed according to the current recommendations [34]. We detected the tetraspanin-specific marker CD63, as well as the escort complex associated markers Tsg101 and Syntenin, in all EV preparations. Varying amounts might be due to the different composition of EV sources, which is in concordance with previous studies [6,21]. Cellular contamination was excluded by the intercellular protein cytochrome C, which was not detected in the EV preparations.

Although a clear discrimination between physiological and pathological EV serum levels could be obtained, increased EV levels were independent of the FIGO staging or the CTC status and, thus, cannot be used to discriminate the clinical profile of EOC patients.

Stratification of EV regarding their HLA-G content revealed that HLA-G_EV_ could not be detected at all in some healthy donors, whereas levels increased with advanced cancer stages in EOC. This suggests an association between the amount of HLA-G_EV_ and the tumor burden. Neo-ectopic expression of HLA-G as a vesicular form can be considered as a critical factor for cancer progression. Indeed, patients with high levels of HLA-G_EV_ had a reduced PFS, pointing to the notion that HLA-G-bearing EV facilitate or support tumor escape from the immune system. This concept is further supported by the association of HLA-G and CTC status before therapy: Levels of HLA-G_EV_ were elevated in patients positive for CTC, compared to patients without CTC. As CTC are malignant cells released by tumor tissues undergoing epithelial-mesenchymal transition, while EV are released by both malignant and nonmalignant cells, EV signals may originate not only from the tumor-derived EV, but also from EV secreted by other cells, such as immune cells [19]. Similarly, it has already been demonstrated that soluble HLA-G can be derived from different cell sources [37,38]. Of note, CTC themselves exploit a large variety of immune-escape mechanisms, including aberrant expression of immune checkpoint molecules (e.g., FASL or PD-L1) [39]. HLA-G bearing EV derived from renal cancer stem-cells were shown to induce the inhibition of dendritic cell differentiation, thereby modifying immune responses towards tumor cells [40]. Independent of the source of vesicular HLA-G, the presence of HLA-G seems to render (tumor) cells immunologically invisible by transducing inhibitory signals towards effector cells promoting cancer progression. Interestingly, we determined an association between high HLA-G_EV_ levels and MUC or EpCAM positive CTC, both being related to poor clinical outcomes in malignancies [41,42,43]. Thus, presence of CTC and HLA-G_EV_ might represent different manifestations of the same phenomenon, suggesting a more ‘immunosuppressive’ subtype of EOC, which is clinically present with higher stages. Nevertheless, this observation should be validated in a cohort encompassing more CTC positive EOC patients. Concerning immunotherapeutic approaches to eliminate single tumor cells in blood and bone marrow, we have been able to show that the application of the intraperitoneal trifunctional bispecific antibody catumaxumab was successful in advanced EOC [44]. For HLA-G_EV_ it would be desirable to develop therapeutic options, either preventing release of HLA-G bearing EV or neutralizing functionality of HLA-G by, for example, receptor blockade.

Although HLA-G_EV_ status does not have a prognostic relevance for patients resistant to platinum therapy, it allows discrimination of patients responding to platinum therapy: Patients with higher levels of HLA-G_EV_ had an inferior PFS, compared to patients with lower levels, suggesting that HLA-G_EV_ might act as a mediator of tumor recurrence despite initial treatment response.

In addition, besides platinum sensitivity, residual tumor burden after primary debulking surgery is one of the most important prognostic factors in EOC [44]. Nevertheless, despite macroscopic complete tumor resection, patients experienced disease recurrence within three years in our cohort. According to our results, stratification of these patients based on the HLA-G_EV_ status might help to identify patients at risk for disease progression. Indeed, the HLA-G_EV_ status was found to be an independent predictive parameter besides platinum sensitivity and presence of CTC before therapy.

Although our study had several limitations, mostly due to the small sample size and its retrospective nature, the clinical relevance of our findings is especially of interest for patients with a rather good prognosis, as defined by sensitivity to platinum therapy and/or no residual tumor burden. As of now, prediction of post-operative outcome is still difficult and clinical examination, as well as imaging techniques, still do not allow for a sufficient prediction of disease recurrence. Thus, risk assessment with additional parameters and novel individualized therapy concepts are urgently needed for patients with a rather good prognosis. Here, HLA-G_EV_ might be a promising candidate as a biomarker easily accessible through liquid biopsy, guiding the development and establishment of individualized therapy concepts. Nevertheless, further research will have to elucidate the prognostic significance of HLA-G_EV_ status and its possible role as a prognostic and predictive biomarker in EOC.

## 4. Materials and Methods

### 4.1. Patient Characteristics

Our cohort consisted of 78 patients diagnosed with histologically confirmed EOC between 2001 and 2014 at the Department of Gynecology and Obstetrics, University Hospital Essen. Tumor classification followed the WHO classification of tumors of the female genital tract. Grading was conducted using the grading system proposed by Silverberg and tumor staging was classified according to the Fédération Internationale de Gynécology et d’Obstétrique (FIGO). All patients underwent primary radical surgery including abdominal hysterectomy, bilateral salpingo-oophorectomy, infragastric omentectomy, peritoneal stripping, and systematic pelvic and paraaortic lymphadenectomy, if indicated. All patients received at least six cycles of carboplatinum AUC 5 and paclitaxel 175 mg/m^2^. Tumors were defined as platinum-resistant if they recurred within six months after the completion of the adjuvant platinum treatment. Any macroscopic residual disease at the end of primary surgery was defined as ‘residual tumor’. Inclusion criteria were: Histologically confirmed EOC, primary radical surgery, platinum-based chemotherapy, and availability of serum-samples and follow-up information. All patients from the selected time period who met these criteria were included. Chemotherapy was administered post-operatively in all patients during this period. Clinical characteristics of the patients are documented in Table 1. A total of 30 healthy female donors served as controls. Written informed consent was obtained by all participants and the study was approved by the Local Ethics Committee (Essen 05-2870 and 17-7859) and was performed according to the declaration of Helsinki. Patient characteristics at time of primary diagnosis are summarized in Table 1.

### 4.2. Sampling of Blood

At the time of primary diagnosis, prior to surgery, 10 mL ethylenediaminetetraacetic (EDTA) blood was collected for isolation of CTCs before the application of therapeutic substances with an S-Monovette (Sarstedt AG & Co., Nürnbrecht, Germany) and stored at 4 °C until further examination. The samples were processed within 4 h after blood collection.

### 4.3. Selection, Detection and Evaluation of CTCs

AdnaTest OvarianCancer (QIAGEN, Hilden, Germany) was employed for enrichment of CTCs and subsequent expression analysis. This test has been described in detail [4,45].

### 4.4. Isolation of Extracellular Vesicles

Plasmatic extracellular vesicles were isolated using ExoQuick™ (SBI Systems Bioscience Inc., Mountain View, VA, USA), as described previously [21].

### 4.5. EV Characterization by Western Blot

EV characterization by SDS-PAGE and western blot analysis was performed as previously described [21]. ExoQuick™ reagent was removed from EV suspensions by PD SpinTap G-25 (GE Healthcare, Freiburg, Germany) prior to SDS-PAGE and western blotting. EV suspensions (15 µg) from four healthy controls and from four EOC patients were analyzed. Cell lysate derived from HEK G1 cells (10 µg) served as control.

### 4.6. Nanoparticle Tracking Analysis

Particle number and size were analyzed using the ZetaView Laser Scattering Video Microscope (Particle Metrix GmbH, Meerbusch, Germany) and its corresponding software (version 8.03.08.02), as previously described [46].

### 4.7. Quantification of Soluble HLA-G Components

Soluble HLA-G was quantified, as previously described [21,47]. EV suspensions were used in a dilution of 1:2 in PBS and purified HLA-G1 [48] served as standard reagent. Soluble HLA-G levels were determined by four-parameter curve fitting. ELISA detection limit of HLA-G was 0.25 ng/mL. HLA-G concentration in EV fractions was considered as HLA-G_EV_.

### 4.8. Statistical Analysis

Statistical analysis was performed using SPSS 22.0 (SPSS Inc., Chicago, IL, USA) and the GraphPad Prism V6.0 software (GraphPad Software, San Diego, CA, USA). Except for particle size (mean ± SD), all metric parameters were given as median and range. After testing for Gaussian distribution, data sets were analyzed using either the Mann–Whitney test or the Kruskal–Wallis test with Dunn’s correction for multiple comparison. Receiver operating curve (ROC) analysis was performed to obtain a cut-off value in terms of sensitivity and specificity, using the BIAS 10.02 software program (http://www.biasonline.de/) for categorization of HLA-G_EV_ status. Progression-free survival (PFS) analyses were assessed by the method of Kaplan–Meier and compared using log-rank test, implemented in the R package survminer (version 0.4.0; https://CRAN.R-project.or/package=survminer). Starting points were time point of diagnosis (blood collection) and endpoint was progress or relapse of EOC disease (therapy requirement). Multivariate Cox regression according to proportional hazards assumption was used to assess the risk of progression.

## 5. Conclusions

In conclusion, our study highlights that a certain subset of EV with HLA-G_EV_, but not the abundance of total EV, offer great clinical and prognostic potential in EOC patients. The molecular characterization of these EV will provide substantial information regarding course of disease and may facilitate individual patient risk management with novel therapy concepts in EOC patients; especially in patients with a rather good prognosis, as defined by sensitivity to platinum therapy or no residual tumor burden.

## Figures and Tables

**Figure 1 cancers-11-01106-f001:**
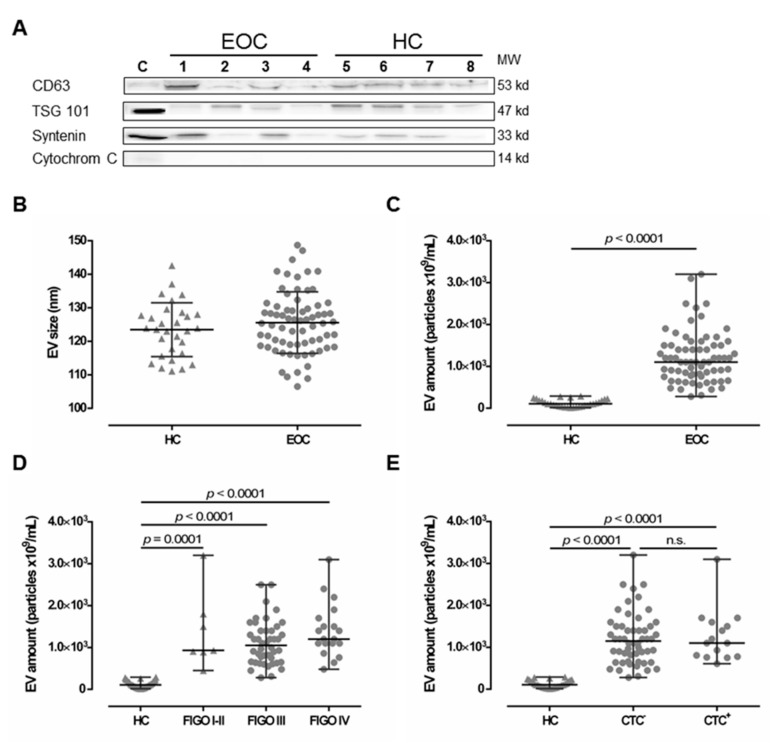
Comparison between extracellular vesicle (EV) preparations derived from plasma of epithelial ovarian cancer (EOC) patients and of healthy controls (HC). (**A**) Representative EV marker expression analysis for CD63, Tsg101, syntenin, and Cytochrome C in EV preparations from EOC patients (1–4) and four HC (5–8). Cell lysate of HEK G1 cells served as control. Detailed information can be found in Appendix A. (**B**) EV size analysis determined by nanoparticle tracking analysis revealed that EV size is similar in both HC and EOC patients. The mean ± SD is given. (**C**) The EV amount was determined by NTA and showed that EOC patients harbor significantly higher amounts of EV, compared to HC. (**D**,**E**) Stratification of EV amount with regards to the International Federation of Gynecology and Obstetrics (FIGO) staging and presence of circulating tumor cells (CTC) showed enhanced EV amounts among all FIGO stages and irrespective of CTC status compared to HC. Given is the median with range. Statistical significance was tested by Mann–Whitney test (**B**,**C**) and Kruskal–Wallis test (**D**,**E**), *p* < 0.05. n.s., not significant. Triangular symbols represent healthy controls, while round circular symbols illustrate EOC patients.

**Figure 2 cancers-11-01106-f002:**
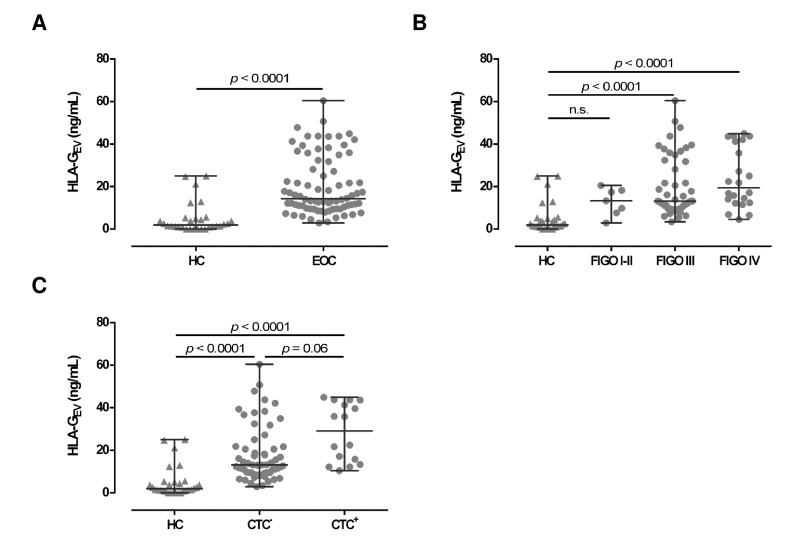
Association of HLA-G_EV_ levels of EOC patients, in comparison to HC. (**A**) HLA-G_EV_ levels are significantly increased in EOC patients, compared to HC. (**B**) Levels of HLA-G_EV_ are increased in advanced FIGO stages. (**C**) EOC patients with detectable CTC harbor higher HLA-G_EV_ levels, compared to HC and patients without CTC. Given is the median with range. Statistical significance was tested by Mann–Whitney test (**A**) and Kruskal–Wallis test (**B**,**C**), *p* < 0.05. n.s., not significant. Triangular symbols represent healthy controls, while round circular symbols illustrate EOC patients.

**Figure 3 cancers-11-01106-f003:**
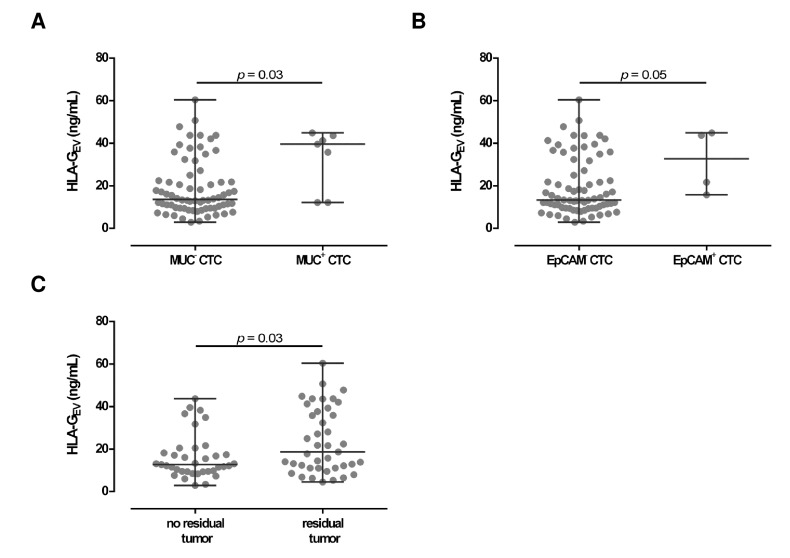
Association between HLA-G_EV_ levels with clinical parameters in EOC patients. (**A**,**B**) Levels of HLA-G_EV_ are elevated in both patients with MUC and EpCAM positive CTC, compared to patients carrying MUC or EpCAM negative CTC. Given is the median [range] in ng/mL. Statistical significance was tested by Mann–Whitney test, *p* < 0.05. (**C**) HLA-G_EV_ levels were higher in EOC patients with a residual tumor burden compared to patients without a detectable residual tumor. Statistical significance was tested by Mann–Whitney test, *p* < 0.05.

**Figure 4 cancers-11-01106-f004:**
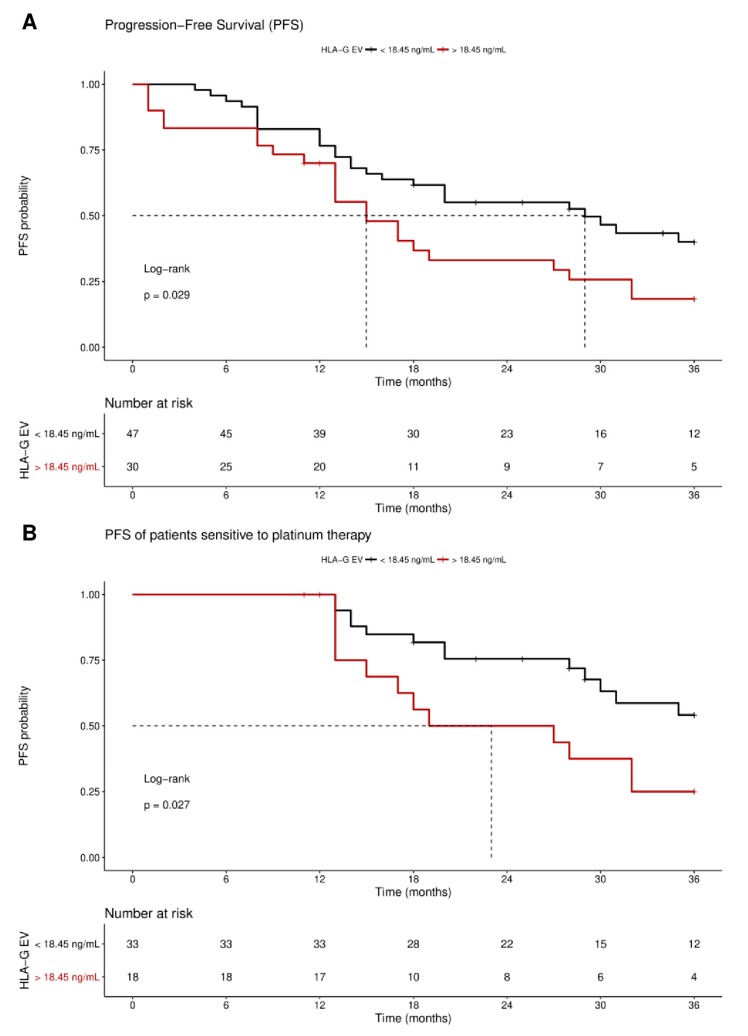
Kaplan–Meier survival analysis regarding vesicular-bound HLA-G. Kaplan–Meier analysis combined with log-rank test revealed that high levels of HLA-G_EV_ are associated (**A**) with a significantly reduced 3-year PFS, and (**B**) with an inferior 3-year PFS in patients sensitive to platinum therapy. The red line indicates patients with HLA-G_EV_ levels greater than 18.45 ng/mL, whereas the black line illustrates patients with HLA-G_EV_ levels less than 18.45 ng/mL. The dotted lines reveal the median survival time, where applicable. Tables under Kaplan–Meier plots show the corresponding numbers at risk.

**Figure 5 cancers-11-01106-f005:**
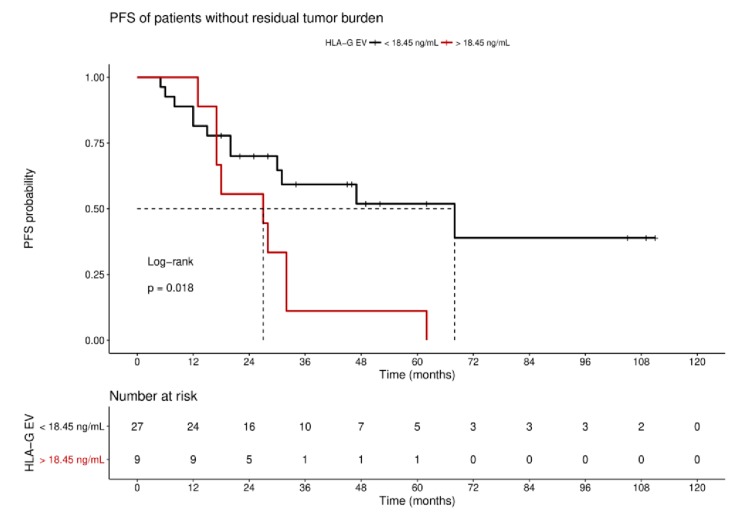
Kaplan–Meier survival analysis regarding HLA-G_EV_ status for patients without residual tumor burden. Kaplan–Meier analysis combined with log-rank test demonstrated that high levels of HLA-G_EV_ are significantly associated with a reduced 10-year PFS in patients without residual tumor burden. The red line represents patients with HLA-G_EV_ levels greater than 18.45 ng/mL, whereas the black line illustrates patients with HLA-G_EV_ levels less than 18.45 ng/mL. The dotted lines indicate median survival time. Tables under Kaplan–Meier plots show the corresponding numbers at risk.

**Table 1 cancers-11-01106-t001:** Patient characteristics at time of primary diagnosis.

Total		Total EV Amount * *n* = 70 (%)	HLA-G_EV_	
			*n* = 78 (%)	Median [Range] ng/mL
Age		Median: 67 (42–98)	Median: 67 (42–98)	
FIGO stage	I–II	7 (10%)	7 (9%)	13.3 [2.9–20.5]
	III	44 (63%)	49 (63%)	13.1 [3.4–60.4]
	IV	19 (27%)	22 (28%)	19.5 [4.5–44.9]
Nodal status	N_0_	16 (23%)	18 (23%)	12.9 [2.9–36.7]
	N_1_	32 (46%)	34 (44%)	13.6 [3.4–44.9]
	unknown	22 (31%)	26 (33%)	
Metastases formation	M_0_	51 (73%)	56 (72%)	13.1 [2.9–60.4]
	M_1_	19 (27%)	22 (28%)	19.5 [4.5–44.9]
Tumor grading	I-II	27 (39%)	30 (38%)	16.3 [4.5–50.7]
	III	43 (61%)	48 (62%)	13.2 [2.9–60.4]
Residual tumor	no	34 (49%)	37 (47%)	12.7 [2.9–43.7]
	yes	36 (51%)	41 (53%)	18.7 [4.5–60.4]
CTC before therapy	negative	54 (77%)	61 (78%)	13.1 [2.9–60.4]
	positive	15 (22%)	16 (21%)	29.1 [10.4–44.9]
	unknown	1 (1%)	1 (1%)	
CTC before therapy	MUC negative	61 (88%)	70 (90%)	13.6 [2.9–60.4]
	MUC positive	8 (11%)	7 (9%)	39.6 [12.2–44.9]
	unknown	1 (1%)	1 (1%)	
CTC before therapy	EpCAM negative	64 (92%)	73 (94%)	13.3 [2.9–60.4]
	EpCAM positive	5 (7%)	4 (5%)	32.7 [15.8–44.9]
	unknown	1 (1%)	1 (1%)	
CTC after therapy	negative	17 (24%)	20 (26%)	12.8 [5.3–47.8]
	positive	7 (10%)	7 (9%)	12.7 [8.0–44.9]
	unknown	46 (66%)	51 (65%)	
DTCs before therapy	negative	40 (57%)	46 (59%)	13.6 [2.9–50.7]
	positive	28 (40%)	30 (38%)	15.0 [6.0–60.4]
	unknown	2 (3%)	2 (3%)	
Platinum-based chemotherapy	no resistance	48 (69%)	51 (66%)	13.3 [2.9–60.4]
	resistance	9 (13%)	12 (15%)	15.5 [5.3–44.9]
	unknown	13 (18%)	15 (19%)	
Recurrence (10y)	no relapse	23 (33%)	24 (31%)	13.2 [3.4–50.7]
	relapse	46 (66%)	53 (68%)	16.8 [2.9–60.4]
	unknown	1 (1%)	1 (1%)	
Overall Survival (10y)	alive	38 (54%)	41 (53%)	13.1 [2.9–60.40]
	dead	31 (45%)	36 (46%)	17.0 [4.5–44.9]
	unknown	1 (1%)	1 (1%)	

CTC—circulating tumor cell; DTCs—disseminated tumor cells; FIGO—Federation of Gynecology and Obstetrics; M_0_—no metastasis formation; M_1_—metastasis formation; pN_0_—no nodal infestation; pN_1_—nodal infestation; * EV particle could not be obtained for all patients.

**Table 2 cancers-11-01106-t002:** Multivariate analysis to predict disease progression in EOC patients.

Risk Factors		*n*	3-Year PFS		10-Year PFS	
			*p*	HR (95% CI)	*p*	HR (95% CI)
Metastasis	M_0_	34	0.467	1.7 (0.4–7.7)	0.351	2.0 (0.5–9.0)
	M_1_	10				
Nodal status	N_0_	16	0.858	1.1 (0.4–3.2)	0.819	0.9 (0.3–2.4)
	N_1_	28				
CTC before therapy	negative	36	0.101	2.8 (0.8–9.7)	0.08	3.0 (0.9–10.0)
	positive	8				
Residual tumor burden	yes	13	0.023	2.9 (1.2–7.4)	0.011	3.3 (1.3–8.4)
	no	31				
Platinum-based chemotherapy	no resistance	36	<0.0001	16.4 (4.5–59.2)	<0.0001	28.8 (6.4–130.0)
	resistance	8				
HLA-G_EV_ status	<18.45 ng/mL	33	0.029	2.9 (1.1–7.6)	0.006	3.8 (1.5–9.9)
	>18.45 ng/mL	11				

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
