# Peer review of "Vesicular-Bound HLA-G as a Predictive Marker for Disease Progression in Epithelial Ovarian Cancer"

_cancers, 2019, doi:10.3390/cancers11081106_

Round 1

Reviewer 1 Report

Line 25 abs CTC please explain acronym Circulating tumor cells

Line 334 : 15 μâ—‹g of EV… Please check typingl error. It should be… 15 μg of EV

Author Response

Thanks for the positive rating of our manuscript.

Reviewer 2 Report

Very concise written paper with an elegant method of detecting biomarkers: HLA-Gev, CTCs and EV, the authors draw the right conclusions. HLA-GEV identifies patients with high risk of Progression - methodes fine with well taken cotrol Group. Further Evaluation of this marker is warranted

Author Response

Thank you for your positive feedback.

Reviewer 3 Report

1) In figure 2C, comparing three roups with t-test is not recommended as ONE-Way ANOVA should be used.

2) Line 152, There is a trend (as p=0.06 and more than 0.05 ; also the test is mann-whitney which is not suitable for three groups) for CTC+ patients but perhaps a study with more samples of CTC+ is much needed to prove the author conclusion.

3) Was there any difference in the total sHLA-G levels in plasma and sHLA-G EVs levels, if yes did the total sHLA-G affected findings of sHLA-G+ EVs in any way?

Author Response

Regarding comment 1: 

Thank you for your comments. We performed One-way ANOVA and obtained the same significances by Kruskal Wallis test for multiple comparisons. We changed the legend of Figure 2 accordingly.

Regarding comment 2: 

We included this issue in the discussion (ll 265ff): Nevertheless, this observation should be validated in a cohort encompassing more CTC positive EOC patients.

Regarding comment 3: 

Thank you for your comment. We previously showed that total amounts of sHLA-G are elevated in EOC compared to healthy controls (Schwich, 2019, Sci Rep, doi.org/10.1038/s41598-019-41900-z), but no associations with the clinical status or outcome were found. We implemented this issue already in the introduction (ll. 81ff), and added this issue also in the discussion (ll. 223ff): Although, the immune checkpoint molecule HLA-G has been associated with cancer invasiveness and metastatic spread in various malignant diseases [35, 36], we were not able to correlate the total amount of sHLA-G to the clinical status or outcome of EOC patients in a previous study [33].

Round 2

Reviewer 3 Report

I wanted to know if there is additional soluble component of HLA-G other than EVs. but I think they havent done this on full blood so it wont be possible for them to answer this.